# A Novel View of Human *Helicobacter pylori* Infections: Interplay between Microbiota and Beta-Defensins

**DOI:** 10.3390/biom9060237

**Published:** 2019-06-18

**Authors:** Raffaela Pero, Mariarita Brancaccio, Sonia Laneri, Margherita-Gabriella De Biasi, Barbara Lombardo, Olga Scudiero

**Affiliations:** 1Dipartimento di Medicina Molecolare e Biotecnologie Mediche, Università degli Studi di Napoli “Federico II”, 80131 Napoli, Italy; barbara.lombardo@unina.it; 2Task Force sugli Studi del Microbioma, Università degli Studi di Napoli “Federico II”, 80131 Napoli, Italy; 3Department of Biology and Evolution of Marine Organisms, Stazione Zoologica Anton Dohrn, 80121 Napoli, Italy; mariarita.brancaccio@szn.it; 4Dipartimento di Farmacia, Università degli Studi di Napoli “Federico II”, Via Montesano 49, 80131 Napoli, Italy; slaneri@unina.it (S.L.); margherita.debiasi@unina.it (M.-G.D.B.); 5CEINGE-Biotecnologie Avanzate Scarl, Via G. Salvatore 486, 80145 Napoli, Italy

**Keywords:** *Helicobacter pylori*, defensins, microbiome, infections

## Abstract

The gut microbiota is significantly involved in the preservation of the immune system of the host, protecting it against the pathogenic bacteria of the stomach. The correlation between gut microbiota and the host response supports human gastric homeostasis. Gut microbes may be shifted in *Helicobacter pylori* (*Hp*)-infected individuals to advance gastric inflammation and distinguished diseases. Particularly interesting is the establishment of cooperation between gut microbiota and antimicrobial peptides (AMPs) of the host in the gastrointestinal tract. AMPs have great importance in the innate immune reactions to *Hp* and participate in conservative co-evolution with an intricate microbiome. β-Defensins, a class of short, cationic, arginine-rich proteins belonging to the AMP group, are produced by epithelial and immunological cells. Their expression is enhanced during *Hp* infection. In this review, we discuss the impact of the gut microbiome on the host response, with particular regard to β-defensins in *Hp*-associated infections. In microbial infections, mostly in precancerous lesions induced by *Hp* infection, these modifications could lead to different outcomes.

## 1. Introduction

*Helicobacter pylori* (*Hp*) infections are related to the onset of various gut diseases, such as gastritis, gastric cancer, and mucosa-associated lymphoid tissue lymphoma (MALT) [1,2,3]. Different studies have been developed to understand how *Hp* can cause local and systemic effects on the human host. An important mechanism involved in these effects is probably injury to the host as a result of chronic inflammation [4]. Recently, various studies have started to concentrate on the impact of *Hp* and its metabolism on the gut microbiome [5,6,7,8]. This rising area could partly elucidate the large heterogeneity of results that are, at present, due to the infection of *Hp* in the host.

This review describes the effects of the gastrointestinal microbiome on the host response, with particular regard to the role of β-defensins in *Hp*-related infections.

## 2. Human *Hp* Infections

*Hp* infection is acquired throughout babyhood through intrafamilial transmission, and in many cases, it proceeds, unless removed by antibiotic treatment [9]. Chronic infection by *Hp* results in mucosal gastric inflammation, which is devoid of clinical symptoms in most infected subjects. Only a minority of infected people develop severe gastroduodenal diseases [10]. Among these infected individuals, about 10% develops ulcers, 1–3% develop gastric carcinoma, and fewer than 0.1% develop gastric MALT (Associated Lymphoid Tissue Lymphoma). 

Intestinal epithelial adenocarcinoma is the most common type of *Hp*-induced gastric cancer. This type of cancer begins with passage from the normal mucous membrane to chronic gastritis. Progression through a series of distinct histologic steps from atrophic gastritis is succeeded by enteral metaplasia, resulting in dysplasia and adenocarcinoma [11,12].

Several other factors are involved in the cancerogenesis process such as epigenetic [13,14] and inflammatory [15] alterations triggered by *Hp* in the host cell genome. These represent critical hallmarks of gastric cancer. 

Gastritis generally consists of inflammation of the mucosal lining of the stomach, which can subsequently lead to the development of ulcers. The ruling etiology of gastritis worldwide is thought to be *Hp* infection, and *Hp* infection also augments the risk of non-cardia gastric cancer by six to eightfold [16,17].

Gastritis induced by *Hp* can affect antral or corpus gastric function. In antrum gastritis, *Hp* provokes an increase in gastrin secretion which leads to greater production of gastric acid, which renders subjects more susceptible to peptic ulcers, but less predisposed to gastric cancer (GC). In corpus gastritis, *Hp* inhibits the production of acid via inflammation, which causes a gradual leak of gastric glands and finally, leads to atrophic gastritis [16].

The decreased secretion of gastric acid promotes the persistence of bacteria usually killed by the adverse environment of the stomach. 

The way in which the altered microbiota interacts with *Hp* to prompt tumorigenesis is not completely known. Probably, these microorganisms can transform nitrogen compounds into carcinogenic N-nitroso compounds.

For example, *Lactobacillus*, *Escherichia coli*, and *Staphylococcus*, can produce N-nitroso compounds [18,19]. On the contrary, *Streptococcus*, *Prevotella*, and *Neisseria* commensals are related to a poor risk of development of gastric cancer [20,21] (Table 1).

## 3. Control Human Gut Microbiome

The microbiome represents the whole microbial community present in the body of a defined host. In humans, the microbiome contains bacteria, viruses, fungi, and Archeae [22,23,24]. The microbiome mutates profoundly among subjects, and colonization by particular microbes could confer protection or susceptibility to disease, especially in chronic infections related to *Hp* [25].

The majority of the human gut microbiome is represented by anaerobic bacteria, such as *Bacteroidetes*, *Firmicutes* or *Proteobacteria*, while a minor proportion (<1% in frequency) belong to other phyla, such as *Actinobacteriae*, *Acidobacteriae*, *Verrumicrobiae*, or *Fusobacteriae*. Moreover, inside a specific level of the GIT (gastrointestinal tract), we can also observe a difference in bacterial composition (Figure 1).

To this level-specific variation, diversity between the epithelium and the intestinal lumen is added.

This diversity is caused by the production of mucus by the goblet cells, limiting the capability to adhere and invade the specialized bacteria [26,27,28].

The glycosylated proteins of the mucus represent a nutrient tank for bacteria such as *Clostridium*, *Lactobacillus*, or *Enterocuccus*, which is used before accessing the intestinal cells [29]. Differences have been shown between diversity and compositions of biopsies or stools [30].

Also, in the microbiome of the gastric corpus, the species richness is higher than in the antrum; however, no differences have been found regarding diversity.

Furthermore, in the gastric corpus, a more diversified microbiome is present in *Hp*-negative individuals compared to *Hp*-positive individuals. An analysis of beta diversity showed that the status of *Hp* infection is more significant than the antrum or corpus [31]. 

In addition, an analysis of the gastric microbiota showed comparable settlement in control individuals from diverse cultural groups and different geographic backgrounds [32].

## 4. *Hp* and the Gut Microbiome

*Hp* can cause alterations in the host by modifying the gut microbiome. A growing number of studies is reporting greater ecosystem diversity in the gastrointestinal tract and associating the presence of *Hp* with variations within the structure of the microbiome [33,34,35,36,37,38,39]. 

The most common genera in healthy and negative-*Hp* gastritis subjects are *Neisseria*, *Prevotella Porphyromonas*, and *Haemophilus* [29]. Additionally, *Lactobacillus*, *Streptococcus*, and *Propionibacterium* represent prevailing genera in normal people [36].

At the phylum level, the presence of Hp has no result on the diversity or uniformity of the gastrointestinal microbiome [35]. 

However, the presence of *Hp* is able to induce drastic alterations in the variety of gut microbiota [6,39,40].

Modifications due to the presence of *Hp* concern the raise in the relative richness of *Acidobacteria*, *Spirochetes*, and *Proteobacteria*, and the reduction of *Firmicutes*, *Actinobacteria*, and Bacteroidetes [39]. Besides, greater amounts of *Firmicutes*, *Actinobacteria* and *Bacteroidetes* have been reported in people with low *Hp* levels. Such divergences could be due to variations between subjects, as the gut microbiome appears to be responsive to external influences, such as lifestyle and nutrition [41,42].

The time period for acquiring *Hp* infection is another aspect to consider in the divergence reported for the microbiome variations associated with *Hp*. Modifications in the variety and constitution of the community have been ascertained in the guts of children and even in relation to adults, despite the *Hp* status [43]. 

Therefore, the premature acquisition of bacteria probably forms the structure of the microbiome through the induction of native changes in the gastric habitat. A mechanism involved in these effects could be the output of ammonia and bicarbonate, starting by urea. Ammonia and bicarbonate might function as substrates for alternative bacteria, as well as changing the pH of the stomach, which promotes the settlement of alternative species, such as nitrogen-reducing microbes [44,45,46,47].

The effect of *Hp* on acid secretion looks to depend upon the type of *Hp*-induced gastritis. The production of gastric acid increases in antral gastritis and decreases in that of the body. Therefore, changes to the microbiome could be problematic in each case. Hyperchlorhydria will increase the microbial variety in the gastric tract and has been related to the development of gastric carcinogenesis. In addition, the consistency of gastric mucus decreases with an increasing *pH*, making it more accessible to alternative bacteria to colonize epithelial tissue. Finally, *Hp* may directly modify the mucosal barrier by altering the expression of gastric mucins [48,49,50,51,52,53,54]. 

These ambient alterations in the gut may affect the natal microbiome [55]. Additionally, the immune reaction driven by *Hp* may alter the native microbiome further, as microorganism populations exist at a lot of distal sites in the human organism. 

In Mongolian gerbils, chronic *Hp* infection causes degenerative alterations. These alterations occur with variation in the expression of genes linked to immunity in both the stomach and respiratory organs [56]. Perhaps, premature settlement changes could lead to alterations in the hollow microenvironment, while late ones could be conducted by an additional shift in the inflammatory and immune response primed by *Hp* [57]. It is possible that *Hp* may have an effect on mucosal diseases in remote sites through its impact on immune cells that move through the body’s circulation. *Hp* even provokes a change in the immune reaction through the elicitation of Treg cells (regulatory T cells) with VacA (vacuolating cytotoxin A) and GGT (gamma-glutamyl transpeptidase) intermediates [58,59]. Treg cell reactions are critical for the discrimination between self and foreign antigens, i.e., immune tolerance. This discovery may explain the persistence of *Hp* in the stomach and could also help suppress gastric inflammation, which may justify a reduction in the severity of gastric disease in *Hp*-positive children related to *Hp*-positive adults [42]. The infection by *Hp* in the stomach has been correlated with modifications of the gastrointestinal microbiome at alternative places in the organism. Reasonably, alternative members of gut microbiota can also contribute to changes discovered after *Hp* infection by the host. In fact, there is a direct interaction between *Hp* and *Streptococcus mitis* that causes the passage of *Hp* from a spiral to coccoid shape that is most refractory to stressful environments [60]. In addition, antimicrobial molecules made by *Lactobacillus* spp. are active against *Hp* strains [61,62,63]. Therefore, there is a two-way communication and modulation between *Hp* and other constituents of the microbiota and, consequently, the set of these influences will be reflected by the health of the host [64]. In this way, changes in the microbiome triggered by the first acquisition of *Hp* could also confirm the immunological status of the host and, as a result, the occurrence of many general diseases. This is relevant, since once the microbiota is altered by treatment with antibiotics, *Hp*-induced inflammation is reduced [42].

## 5. Gut Microbiota in *Hp* Infections

The host’s metabolism is affected by *Hp* and gut microbiota. The impact of *Hp* infection on modifications to the gastro-intestinal microbiota has been examined in animals and humans [65,66,67]. 

In rhesus macaques, there are no important changes in the richness of non-*Helicobacter* taxa in the stomach pre-and post-infection with *Hp* [65]. Other studies have shown increases in *Staphylococcus aureus* and *Enterococcus* spp. in the gastrointestinal tract and a down-shift of *Lactobacillus* spp. in the gut of Mongolian gerbils infected with *Hp* [66]. Similarly, mice infected with immunopathological *Hp* B8 can augment the intestinal charges of *E. coli* in the cecum and *Bacteroides/Prevotella* spp. in the colon [60].

*Hp* seems to be a crucial constituent of the gut microbiota and, when present, has the highest relative abundance [35]. A decrease in *Proteobacteria* and an augment in *Firmicutes* have been demonstrated in subjects with *Hp*-positive antral gastritis related to *Hp*-negative gastritis [38]. Similarly, an increase in *Streptococcus* and a reduction in *Prevotella* have been discovered in subjects with atrophic gastritis likened to control ones [32].

The predominance of non-*Hp* microbes was found to be greater in subjects with non-ulcerative dyspepsia than in those with peptic ulcers. Instead, the isolation of *Streptococci* and not of *Hp* has been correlated with the development of peptic ulcers [68,69].

## 6. Gut Microbiome Modifiers

As well as *Hp* presence, the structure of the gut microbiota could be altered by diet, lifestyle, medication use, age, exercise, and genetic factors [68,69,70,71,72,73,74] (Figure 2).

### 6.1. Diet

Dietary factors considerably change both the function and the composition of gut microbiota. A landmark study investigated the microbiota of Italian and countryside African children. Prominent exhaustion of *Enterobacteriaceae* and *Firmicutes* and, remarkably, more short chain fatty acids (SCFAs) were found to occur in African children.

Foods of animal origin were shown to promote the growth of bile-tolerant bacteria microorganisms, while the ability of *Firmicutes* to process the polysaccharides of vegetal foods was diminished. This study highlights a significative correlation between bile acids, saturated milk fats, and excessive proliferation of some proinflammatory microorganisms such as *Bilophila wadsworthia* [75,76,77,78], which is associated with chronic inflammatory diseases [76]. Substantial shifts in the microbial constitution of the intestines were observed during the change to a high fat diet (HFD) with a reduction in the phylum *Bacteroidetes* and increases in the phyla *Firmicutes* and *Proteobacteria*. Furthermore, the consumption of red meat was correlated not only with a poor grade of inflammation, but also with a particular microbiome in subjects with late adenoma and colorectal carcinoma (CRC) [77]. 

### 6.2. Medication Use 

The proton pump inhibitor (PPI) produces a significant decrease in gastric pH, and the constitution of the gut microbiota is more similar to the fecal one than that of the H2 antagonists and the untreated control [79]. Treatment with antibiotics induces modifications in the gastric microbiota and an imbalance between *Bacteroidetes* and *Firmicutes*. The diversity and abundance of these bacteria decrease during antibiotic treatments. The alterations of the microbiome depend on the type of antibiotic used, the dosage, the time of administration, the pharmacological action and the target bacteria. Antimicrobial results or modalities of functioning are crucial for the sorting of gut microbiota and are a reliable way of assessing the composition of microbial alterations during antibiotic treatment [80,81,82].

### 6.3. Age

With age, the microbiota diversity increases until it becomes a stable adult microbiota. Three bacterial phyla dominate the composition, *Firmicutes*, *Bacteroidetes* and *Actinobacteria*, which is influenced by genetic factors, diet, environment, lifestyle, and gut physiology. 

In children, the gut microbiota composition and diversity are like those of adults by about three years of age [83,84].

### 6.4. Exercise

In young children and adolescents, daily exercise increases gut microbial diversity with the enrichment of *Firmicutes*, such as *Clostridiales*, *Roseburia*, *Lachnospiraceae*, and *Erysipelotrichaceae*. These microbes generate SCFAs that can enhance the levels of tight junctions in the gastrointestinal tract to improve colon barrier solidity, reduce mucosa permeability, and inhibit inflammation triggered by cytokines [84,85].

In athletes, higher alpha diversity of gut microorganisms has been demonstrated compared with high and low body mass index (BMI) controls. Indeed, this study showed that exercise elevates gut microbial diversity and protein depletion. Therefore, there is a beneficial impact of activity on the variations of gut microbiota, and the correlation between physical activity and microbiota diversity is probably related to the concomitant dietary extremes [86].

### 6.5. Genetic Factors 

Genetic aspects of the host are able to affect the gastrointestinal microbiota and its metabolic phenotype. 

Allele-specific single nucleotide polymorphisms (SNPs) are related to gut microbiota and are connected with illnesses such as obesity, Type 2 diabetes (T2D), and CRC [87].

In healthy individuals, some fecal bacterial taxa are heritable, and SNPs are associated with *Faecalibacterium*, *Lachnospira*, *Rikenellaceae*, and *Eubacterium* [88].

A gender variance in gut microbiota was also detected owing to differences in daily and social practices [89].

The microbial constitution of the stomach is also influenced by the presence, in the mucosa, of ABH antigen, whose production depends on the *FUT2* (Fucosyltransferase 2) gene. The ABH antigen is one of the most famous polymorphisms in the field of blood antigens in body excretions. On red blood cells, they are in the form of fat-soluble glycolipids, and in secretions, they occur as water-soluble glycoproteins.

Mutations in the *FUT2* gene render individuals unable to secrete the ABH antigen. In these individuals, *Bifidobacterial* variance and abundance, particularly *B. catenulatum/pseudocatenulatum*, *B. adolescentis*, and *B. bifidum*, decreases [90]. 

A very recent study suggested that the protein P1454 is released by *Hp* in infected patients, and it drives Th1 (T cell helper 1) and Th17 (T cell helper 17) inflammatory responses during chronic *Hp* infection and in patients with distal adenocarcinoma [91].

Also, ambient factors predominate above genetics in forming the gut microbiota. In fact, around 20% of the interpersonal variability of the microbiome is related to factors such as lifestyle, anthropometric criteria, diet, and drugs (Figure 3). 

In contrast, in twins, only a small percentage (about 2–8%) of the microbiome taxa is heritable [92].

## 7. Beta-Defensins and Microbiota in *Hp* Infections

### 7.1. Human Defensins: Classification 

Defensins are short proteins belonging to the mammalian antimicrobial peptides (AMPs) family. These proteins have a large range of antimicrobial activity against bacterial cells, viruses, fungi, and protozoa [93,94,95,96].

The primary sequences of defensin consist of preserved cysteine residues, which form disulfide bridges that give rise to a three-dimensional conformation preserved within family members [92]. Defensins fall into three main groups, called α, β, and θ.

α-defensins, including human α-defensin 5 (HD-5) and 6 (HD-6) in humans, are mainly expressed in the Paneth cells of the small intestine [94]. 

In contrast, β-defensins are secreted in the small and large intestines [97]. The expression of θ-defensins is limited to rhesus macaques [98].

The structural features of defensins provide information on their mechanism of action against bacteria. The HD-5 has a three-dimensional dimeric structure, a basket shape conformation, and a typical amphipathic disposition [99]. HD-6 dimers are able to self-associate, creating higher-order oligomers and building lengthy fibers [100]. α-defensins can give rise to membrane pores or canals that lead to ions [101].

Because α-defensins are fixed by three disulfide bonds which confer conformational rigidity, the development of a secondary structure based on the lipid bond is limited [102].

The antimicrobial activity of some defensins is also based on mechanisms different from a straight microbial kill. For example, HD-6 oligomerizes and forms lengthy fibers that serve to trap microbes on the intestinal surface, preventing their translocation through the epithelial barrier.

However, in a reducing environment, the disulfide bonds of HD-6 are reduced, and antimicrobial activity occurs, causing the destruction of the cell wall [103]. Probably, both mechanisms explain the activity of HD-6 against in vivo bacteria. The expression of α-defensins is controlled by the WNT pathway transcription factor TCF4, but their production is independent of microbial signals [104].

Some β-defensins are expressed separately to the microbiota, while other members, including β-defensin 2 (HBD-2), exhibit greater expression in the presence of bacteria [97].

AMPs are induced only as required, thus reducing their probability of unnecessarily altering the composition of the intestinal microbiota or compromising its beneficial contributions.

The harmful actions of AMPs are usually inhibited during storage within vesicles of secretion; otherwise, they may also damage the membranes of host cells [105].

α-defensins are saved as inactive pro-peptides within the secretory granules of epithelial cells. Human pro-α-defensins undergo proteolysis to generate fully developed proteins with bactericidal activity [106].

Filamentation represents yet another mechanism that regulates AMP action. AMPs are also required to limit the entrance of the microbiota to host cells. The intestinal surface is clothed by mucus that behaves like a physical barricade against lumen bacteria and concentrates AMPs near the surface [107].

In the gastrointestinal tract, and probably somewhere else, HBD-1 expression is also relatively controlled by the receptor activated by the nuclear receptor peroxisome proliferator gamma (PPARγ), a crucial factor for gut homeostasis in response to inflammation, diet, and the microbiota [108].

PPARs are elements of the nuclear receptor superfamily and include three elements: PPARα, PPARβ/δ, and PPARγ, also known as nuclear receptor subfamily 1, group C, and members (NR1C)1, 2, and 3 [109,110].

Although there is a spatial barrier between the PPARs and tissues of our body, they form interfaces with each other. The metabolites generated inside the microbiota are assimilated by inflammatory and intestinal cells. These metabolites are carried to various organs through the systemic circulation, where they function as PPAR ligands. Once activated, PPARs modulate: (i) the intestine, (ii) the body’s immune response, and (iii) carbohydrate and fat metabolism. For example, sodium colitis mice dextran sulfate treated with *Lactobacillus paracasei* B21060 upregulates both PPARγ and β-defensin. This enhancement is linked to the renovation of intestinal stability, suggesting that the microbiota modulate PPARγwhile preserving intestinal homeostasis [111].

Additionally, PPARs activate HBD-1 mediated immunity in Crohn’s disease, which constitutes another intestinal anti-inflammatory mechanism [108].

### 7.2. Beta-Defensins in Hp Infections 

Defensins have an essential role in the onset of *Hp*-infections. The distribution of the resulting gastritis is influenced by genetic factors, bacterial virulence, the age of infection, and the environment [112]. *Hp* infection drives a more important increase in HBD-2 than non-*Hp* gastritis [113]. Furthermore, *Hp* induces endogenous HBD-2 expression from gastric epithelial cells, and this improvement is regulated by the recognition receptor NOD1 (Nucleotide Binding Ligand Domain 1) [114]. In *Hp*-induced gastritis, there are essential increases in HBD-1 and HBD-2 expression with respect to the control. The same increases in HBD1 and HBD-2 expression were detected during bacterial inflammation, suggesting an essential role of innate host defense against potentially injurious stimuli in the stomach [115]. Moreover, subjects with *Hp*-infected gastritis overexpresses HBD-2 mRNA, while those who are *Hp*-negative show only weak HBD-2 expression, indicating that the presence of *Hp* enhances HBD-2 expression in the gut. Besides, there is a marked diversity in HBD-2 expression in the gastric corpus of *Hp* individuals related to those who are *Hp*-negative. In addition, HBD-1 is expressed unevenly, further indicating the failure of transcriptional regulators for pro-inflammatory markers in the *HBD-1* gene [116]. 

HBD3 is expressed regularly, despite colonization of *Hp*. Furthermore, HBD-3 is mainly expressed in the gut in the presence of *Hp*, but not in the absence of it [117]. 

HBD-4 is poorly expressed in gastric cells and is mostly up-regulated in *Hp* and non-*Hp* gastritis. The *cagA*^†^ (cytotoxin-associated gene) *Hp* strain results in significant expression of HBD-4 compared with the *cagA*^−^ strain. The pathway of toll-like receptors (TLRs) is not related to modifications in HBD-4 expression; however, p38 mitogen-activated protein kinase plays an essential role [118]. This research shows that *Hp* modulates the expression of β-defensins, which are critical regulators of the innate immune response in the host. Antimicrobial proteins are capable of changing the structure of gut microbiota during host infection. 

There is a robust correlation between *Hp* infection and peptic ulcers. Peptic ulcers represent gaps in the covering of the gastrointestinal mucosa that are usually produced by *Hp* and non-steroidal anti-inflammatory medicaments. A peptic ulcer is linked with significative mortality and complications such as bleeding and perforation. Individuals with a peptic ulcer have less HBD-1 and greater HBD-2 expression than those with a healthy stomach, and this model is particularly marked in individuals infected with *Hp*. Probably, this increased expression of HBD-2 constitutes a protective reaction by the gastric epithelium to restrict the infection [119].

### 7.3. Beta-Defensins and Microbiota

AMPs have a notable impact on the constitution of the gastrointestinal microbiota. The power to elude them has given some pathogens selective supremacy [120,121]. This is especially evident in the case of *Hp*, which uses the cholesterol of the host to obtain resistance against catelicidin-37 (LL-37) [122].

Furthermore, even if the gastritis that promotes *Hp* induces HBD-2, it seems to selectively inhibit another β-defensin, HBD-3 [123].

β-defensins participate in the development of generic biochemical characteristics such as a positive charge and disulfide bonds, which are essential for antibacterial activity [124]. Although these characteristics are rather comparable among the respective HBDs, there may be large variations in their efficacy and type of activity. A suggestive example is HBD-1, which, in vitro, has only a faint antimicrobial function related to HBD-2 or other defensins. The justification for this loss of capacity is that the rather intense antimicrobial activity of HBD-1 is influenced by biochemical activation in reducing conditions [115]. 

Previous studies have provided evidence that the gastrointestinal microbiota modulates the secretion of defensins. The pre-incubation of Caco-2 cells with live *E. faecium* was found to remarkably decrease the internalization of *S. typhimurium*, whereas pretreatment with thermally killed *E. faecium* did not affect the internalization of pathogens [125]. Moreover, only live gut microbiota can actively upregulate the levels of HBD-2 [126,127,128]. In addition, a supernatant of *Hp* bacteria was not sufficient to enhance HBD-2 in MKN45 cells, suggesting a strain-dependent mechanism, and for certain bacteria, a straight interplay between bacteria is essential for the expression of AMPs. Moreover, genetical aspects are involved in the modulation of β-defensins, because it was found that only *cag* pathogenicity island (PAI)-positive *Hp* strains are capable of inducing HBD-2 expression [125]. The redox state of some defensins could be involved in the modulation of the constitution of the microbiota on the surface of the epithelium and, probably, in precluding the translocation of the microorganisms. In particular, the non-pathogenic *E. coli* Nissle 1917 strain evokes the most pronounced expression of β--defensin in vitro [126,128]. It is interesting to note that different mutants of *E. coli* Nissle 1917 are major stimulating factors for the secretion of β-defensin [126]. Furthermore, NF-kB mediates the expression of β-defensin induced by *E. coli* Nissle 1917 via MAPK/AP-1 pathways [127] (Figure 4).

However, other data are necessary to validate whether gut microbiota can enhance β-defensin expression to decrease bacteria colonization and regulate homeostasis of the intestine. Some molecules produced by the intestinal microbiota, including ligands of aryl hydrocarbon (AHR) and butyrate, have been shown to promote the secretion of Interleukin 22 (IL-22) by pancreatic lymphoid cells (ILCs), which consequently upregulates the secretion of β-defensin 14 (BD14) by endocrine cells. Thus, dysbiosis of the microbiota and the weak affinity of the aryl hydrocarbon receptor (AHR) allele seem to explain the deficient secretion of mouse β-defensin 14 (mBD14) in the pancreases of NOD mice [125]. Since only live gut microbiota is able to induce the expression of β-defensins, most likely a specified microbiota metabolic function, the closure of paths for the expression of AHR ligands may have the capability to modulate the flow of β-defensins (Figure 5).

The surface of the intestinal epithelium represents a chemical barrier composed of mucus and AMPs, which are provided by a single layer of intestinal epithelial cells (IECs), Paneth cells, and goblet cells. It has been demonstrated that SCFAs improve both elements of the intestinal barrier. Operating across the short chain fatty acid receptor (GPR43), butyrate brings about the production of AMPs such as Regenerative islet-derived protein IIIγ (RegIIIγ) and defensin in IEC [129]. RegIIIγ is a type of lectin C; it is one of several antimicrobial peptides produced by Paneth cells. Other studies have demonstrated a synergistic mechanism between mucus and the SCFAs caused by AMPs. Butyrate not only influences the yield of cathelicidin but can also improve mucin formation leading to an optimum immunological reaction as opposed to the occurrence of amebic colitis [130]. The induction of β-defensins by butyrate is Muc2-independent, because the presence of mucus inhibits the antibacterial action of β-defensin. In fact, the protein encoded by Muc-2 (mucin 2, oligomeric mucus/gel-forming) is secreted and forms an insoluble mucous barrier that protects the gut lumen. Additional research is necessary to highlight the association of mucus with AMPs in relation to SCFAs [131].

The gut microbiota can regulate the production of defensins that rapidly kill or deactivate microorganisms [132]. Evidence shows that, in host protection, defensins collaborate with various mediators, such as cytokines, chemokines, supplements, and more antimicrobial proteins, as well as cellular elements, to produce coordinate protection against invading pathogens [133].

## 8. Conclusions

*Hp* has a powerful impact on both the stomach habitat and the immune status of the host, leading to alterations in the gastrointestinal microbiome at disparate sites in the organism. These changes are greatly implicated in the pathogenicity of *Hp*-associated diseases. AMPs are meaningfully involved in the preservation of the solidity and equilibrium of the gastrointestinal tract. Knowledge of microbial-dependent or -independent factors that influence the expression of AMPs is essential and could lead to new therapeutic approaches to diseases of the gastrointestinal tract (GI) correlated with pathogenic colonization and disequilibrium of the host microbiota. Thus, knowledge of the mechanisms involved in these processes, as well as the *Hp*-induced microbiome changes that may be connected to the progress of illness, will probably be helpful in predicting and avoiding *Hp*-related diseases. After forming a mechanistic understanding of animal models, reports on the microbiome will open up a new area of study for the clinical government.

## Figures and Tables

**Figure 1 biomolecules-09-00237-f001:**
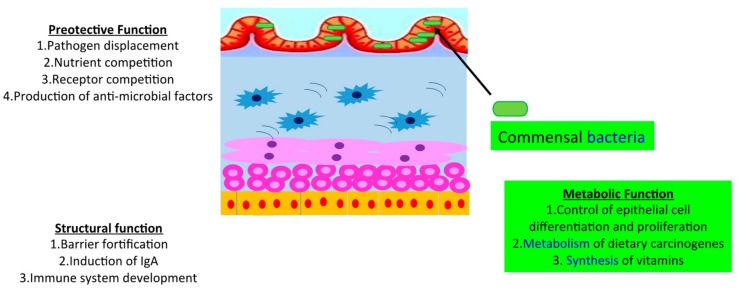
Role of intestinal microbiota.

**Figure 2 biomolecules-09-00237-f002:**
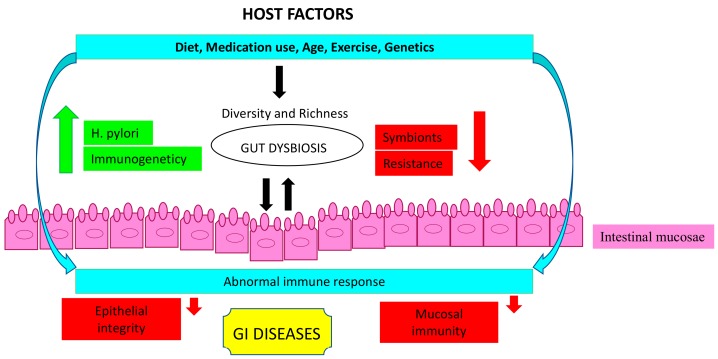
Gut microbiota modifiers. Host factors that modify the gut microbiota composition in the gastrointestinal diseases.

**Figure 3 biomolecules-09-00237-f003:**
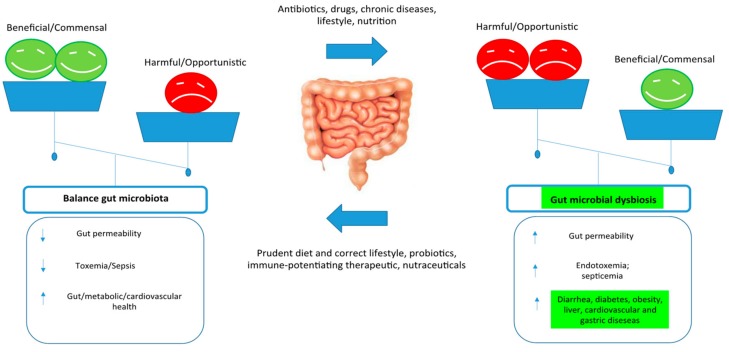
Importance of balanced nutrition and gut microbiota, and consequences of gut dysbiosisis.

**Figure 4 biomolecules-09-00237-f004:**
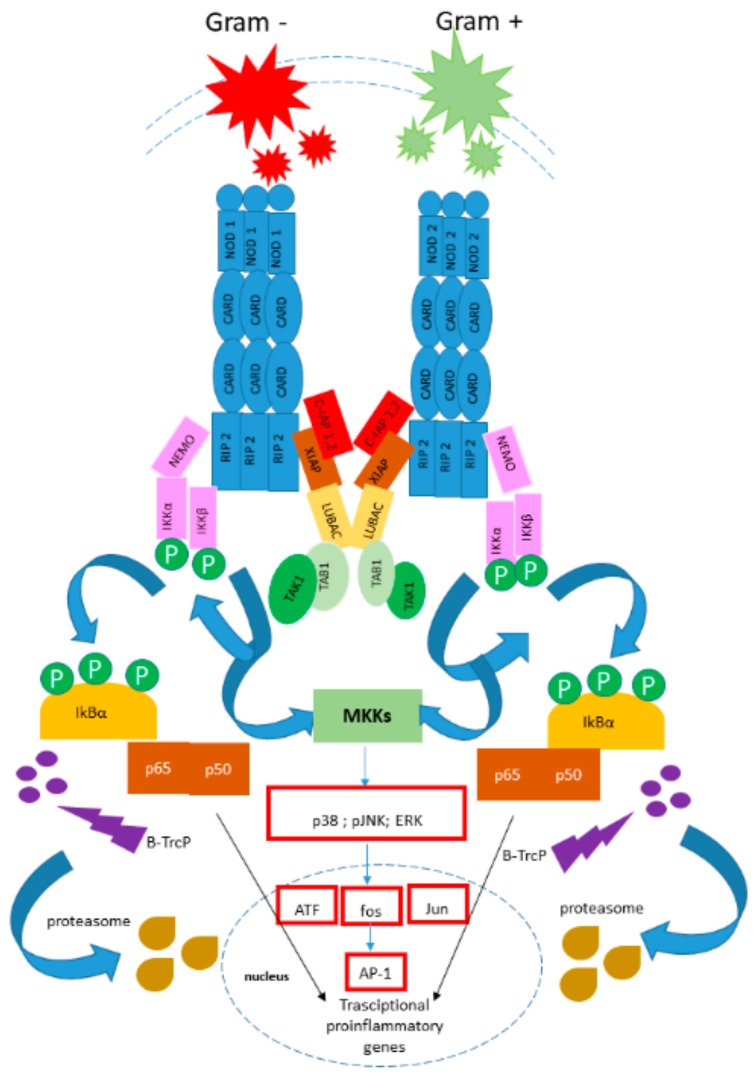
NF-κB and AP-1 pathways. These ligands bind to NOD1 or NOD2 through the LRR domain of these molecules. This interaction initiates the activation of NOD1 and NOD2 due to the induction of a complex conformational change that results in protein oligomerization and further interaction with downstream effectors. NOD1 or NOD2 assembly recruits RIP2 through CARD–CARD interactions, resulting in RIP2 ubiquitination by IAPs and recruitment of the LUBAC complex by XIAP, with further binding of the TAB1/TAK1 complex. It is believed that TAK1 gets activated through autophosphorylation and stimulates the downstream IKK complex, including Lys63-linked polyubiquitination of NEMO (IKKγ), the regulatory subunit of the IKK complex, which also consists of the catalytic subunits IKK1 (IKKα) and IKK2 (IKKβ). This event is followed by IKK2 phosphorylation, which further phosphorylates the NF-κB inhibitor IκBα. IκBα is then ubiquitinated by the SCF/β–TrCP complex and further degraded by proteasome. The degradation of IκBα releases NF-κB dimers to translocate into the nucleus, where they up-regulate target genes involved in host defense and apoptosis. NOD oligomerization and further RIP2 activation also recruits TAB/TAK1 complexes to mediate the phosphorylation of MAPKs, such as JNK, ERK and p38 MAPK, through the upstream activation of MKKs. These kinases translocate to the nucleus and then phosphorylate AP-1 transcription factors (c-fos, c-Jun, ATF and JDP family members) to mediate the expression of target genes containing a TRE (TPA DNA-response element).

**Figure 5 biomolecules-09-00237-f005:**
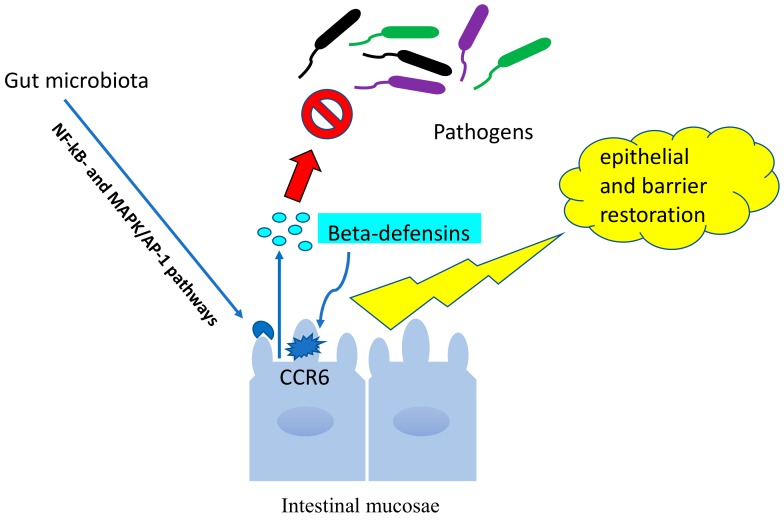
Gut microbiota and induction of human beta-defensins. Gut microbiota secretion via. The NF-kB and MAPK/AP-1 pathways. Beta-defensins interact with CCR6 to restore the epithelium and a barrier.

**Table 1 biomolecules-09-00237-t001:** *Helicobacter pylori* (*Hp*) and human gastric microbiota.

Doi Number Access	Microbiota Changes in *Hp* Patients	Gastric Manifestations in *Hp* Patients
doi: 10.1038/s41598-017-15510-6	*Hp*, when present, tends to predominate, but its abundance may influence and be influenced by the coexisting gastric microbiome. *Hp* disease manifestation may largely be the result of the pathogenicity of its species, and the final outcome may also depend on the network of coexisting microbiota.	Gastritis Peptic ulcer disease Gastric cancer
doi: 10.1111/hel.12293	No significant difference in microbial composition between cancer and control groups under the same *Hp* infection status. An increased proportion of *Actinobacteria* in the cancer groups than in the control groups regardless of *Hp* status was found. In the *Hp* (+) groups, *Staphylococcus epidermidis*, *Klebsiella pneumonia*, and *Neisseria flava* accounted for a larger part of gastric microbiota in the cancer group compared to the control group.	Gastric cancer Dysplasia, Mucosa-associated lymphoid tissue Lymphoma, Esophageal cancer
doi: 10.1097/INF.0000000000001383.	Bacterial richness and diversity of *Hp*-positive specimens were lower than those of negative specimens. *H**p* subjects had a higher relative abundance of the *Helicobacter* genus than *Hp*-negative subjects.	Dyspeptic symptoms
doi: 10.1073/pnas.0506655103	*Hp* does not significantly modify the diversity of the gastric microbiota. There may be geographical variations in the diversity of the gastric microbiome. In contrast, *Hp* influences the microbiota distantly, affecting important target organs	Functional dyspesia Peptic ulcer disease Gastric cancer
doi: 10.1371/journal.pone.0007985	Significantly higher abundance of the *Firmicutes* and *Streptococcus* genus was observed in patients with antral gastritis.	Dyspeptic symptoms
doi: 10.1038/ismej.2010.149	Marked differences were detected in the structure of the gastric bacterial community according to *Hp* status.	Erythematous pre-pyloric region Vesicular lithiasis Antral gastritis Severe inflammation Erosive duodenitis Heartburn/GERD symptoms Dyspepsia
doi: 10.1038/mi.2016.131	The gastric microbiota of *Hp*-infected children was distinct from that of non infected children in terms of the abundance of multiple bacterial classes, orders, families, and genera.	Peptic disease: Dyspepsia Recurrent abdominal discomfort Pain
doi: 10.1056/NEJM199607253350404	Autoimmune and *Hp*-induced atrophic gastritis were associated with different gastric profiles. Proton pump inhibitor (PPI)-treated patients showed relatively few alterations in the gastric microbiota compared to healthy subjects.	Autoimmune atrophic gastritis *Hp*-induced atrophic gastritis *Hp* gastritis
doi: 10.1155/2014/610421	No significant effects on *Hp* or the diversity or composition of human gastric microbiota.	Nonulcer dyspepsia Peptic ulcer disease Gastric cancer
doi: 10.3748/wjg.v18.i11.1257	High prevalence of non-*Hp* bacteria dominated by some species.	Gastritis Gastric ulcer Duodenal ulcer Reflux esophagitis nonulcer dyspepsia

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
