# Peer review of "A Novel View of Human Helicobacter pylori Infections: Interplay between Microbiota and Beta-Defensins"

_biomolecules, 2019, doi:10.3390/biom9060237_

Round 1
Reviewer 1 Report
Pero and colleagues review an interesting topic and suggest a cross-talk between the gut microbiota and the production of beta defensins to explain the contrasting outcomes of Helicobacter pylori infections.
1-I strongly suggest the addition of a table including the reports in which a relation between H pylori infection and changes in the gut microbiota has been observed. A summary of the relevant alterations (functional/compositional) as well as the gastric manifestations of H pylori infection in those patients should also be included.
2-This manuscript has to be reviewed by a native speaker. There are many typos and grammatical errors (including use of verbal tense) in this paper, and as it is some sections in the text are difficult to understand. For citing some, authors have to explain what did they mean with “alterations in breakdown” in line 129. Lines 100-103 are also confusing.
3-In line 71. Avoid the use of expressions such as “healthy microbiome”, since it is not clear if a healthy microbiome can be defined.
Author Response
1) Comment:
I strongly suggest the addition of a table including the reports in which a relation between H pylori infection and changes in the gut microbiota has been observed. A summary of the relevant alterations (functional/compositional) as well as the gastric manifestations of H pylori infection in those patients should also be included.
1) Response: We thank the review for this suggestion. So, we have reported a table (Table 1)in which we describe a summary of the relevant alterations as well as the gastric manifestations of H pylori infection in those patients have been included.
Table 1: Helicobacter pylori (Hp) and human gastric microbiota
Authors. Year |
Microbiota changes in Hp patients |
Gastric manifestations in Hp patients |
Khosravi, et al. 2014 [68] | No significant effects on Hp or the diversity or composition of human gastric microbiota.
| · Nonulcer dyspepsia · Peptic ulcer disease · Gastric cancer
|
Hu, et al. 2012, [69] | High prevalence of the non-Hp bacteria dominated by some species.
| · Gastritis · Gastric ulcer · Duodenal ulcer · Reflux esophagitis · Non-ulcer dyspepsia |
Das, et. al 2017 [6] | Hp, when present, tends to predominate, but its abundance may influence and be influenced by the coexisting gastric microbiome, Hp disease manifestation may largely be the result of the pathogenicity of its species, and the final outcome may also depend on the network of coexisting microbiota. |
· Gastritis · Peptic ulcer disease · Gastric cancer |
Bik, et al. 2006 [6] | Hp does not significantly modify the diversity of the gastric microbiota. There may be geographical variations in the diversity of the gastric microbiome. In contrast, Hp, influences the microbiota distantl, affecting important target organs. | · Functional dyspesia · Peptic Ulcer Disease · Gastric cancer
|
Hyun, et al. 2016 [38]
| No significant difference in microbial composition between cancer and control groups under the same Hp infection status. An increased proportion of Actinobacteria in the cancer groups than in the control groups regardless of Hp status was found. In the Hp (+) groups, Staphylococcus epidermidis, Klebsiella pneumonia, and Neisseria flava accounted for a larger part of gastric microbiota in the cancer group compared to the control group.
| · Gastric cancer · Dysplasia, · Mucosa-associated lymphoid tissue · Lymphoma, · Esophageal cancer |
Llorca, et al. 2017 [33] | Bacterial richness and diversity of Hp-positive specimens were lower than those of negative specimens. Hp subjects had a higher relative abundance of the Helicobacter genus than Hp-negative subjects. | · Dyspeptic symptoms |
Li, et al. 2009 [38] | Significantly higher abundance of the Firmicutes and Streptococcus genus was observed in patients with antral gastritis.
| · Dyspeptic symptoms |
Maldonado-Contreras, et al. 2011 [15] | Marked differences were detected in the structure of the gastric bacterial community according to Hp status.
| · Erythematous pre-pyloric region · Vesicular lithiasis · Antral Gastritis · Severe inflammation · Erosive duodenitis · Heartburn/GERD symptoms Dyspepsia |
Brawner, et al. 2017 [43] | The gastric microbiota of Hp-infected children was distinct from that of noninfected children in terms of the abundance of multiple bacterial classes, orders, families, and genera. | · Peptic disease: · Dyspepsia · Recurrent abdominal discomfort · Pain
|
Parsons, et al. 2018 [48] | Autoimmune and Hp induced atrophic gastritis were associated with different gastric profiles. Proton pomp inhibitors (PPI)-treated patients showed relatively few alterations in the gastric microbiota compared to healthy subjects. | · Autoimmune atrophic gastritis · Hp-induced atrophic gastritis · Hp gastritis |
2) Comment:
This manuscript has to be reviewed by a native speaker. There are many typos and grammatical errors (including use of verbal tense) in this paper, and as it is some sections in the text are difficult to understand. For citing some, authors have to explain what did they mean with “alterations in breakdown” in line 129. Lines 100-103 are also confusing
2) Response: We thanhs the review for the comment. We have used MDPI Editing that provides an English editing service checking grammar, spelling, punctuation and some improvement of style where necessary for an additional charge . We have also clarify the sentences in line 129 and 100-103.
3) Comment: In line 71. Avoid the use of expressions such as “healthy microbiome”, since it is not clear if a healthy microbiome can be defined.
3) Response: we have changed “healthy microbiome” with “control”. (lane 168, pag. 4)
Reviewer 2 Report
The manuscript is quite well written.
It represents a comprehensive review of the topic.
It would be useful for the readers to include the discussion of
PMID:30646431
.
Author Response
Reviewer 2
Response: Thanks to the Reviewers/Editors for his/her appreciation of the manuscript.
1) Comment
Include in the discussion the PMID ref: 30646431
Response: We thanks the reviewers for the suggestion that improves our manuscript. We have included the PMID ref: 30646431 in the manuscript at lane 491-493, pag.8. ref: 93.
Reviewer 3 Report
Dear Authors,
Major comments
This review described the diversity of microbiota under various factors including Helicobacter pylori infection but provides only poor impact for researchers. I think the authors had better reconstruct the contents in order to have further interested to this review. For example, the authors had better present the detailed microbiota altered by various factors using Tables and Figures. In addition, the authors had better indicate the Figures as for the expression mechanism of defensins via the MAPK/AP-1 signaling pathway concerned with the activation of NF-kB.
Minor comments
1. The authors should alter H. Pylori to Helicobacter pylori(Italic) in TITLE.
2. Please rewrite “belongin” to “belonging” in line 22 in ABSTRACT.
3. Please consider to alter “which expression.....” to “ in which the expression of b-defensins.......” in line 23 in ABSTRACT.
4. Please consider to alter “researches” to “studies” in line 35 in INTRODUCTION.
5. Does GC in line 60 mean gastric cancer?
6. What is SCFA in line 178? Is it short-chain fatty acid?
7. What is HFD in line 185?
8. What is CRC in line 188?
9. What is ABH in line 225?
10. What is AMP in line 234?
11. Please check whether there is a hyphen between HBD and number.
12. What are IL-22 and ILCs in line 351?
13. What is BD14 in line 352?
14. What are AHR and mBD14 in line 353?
15. What is CCR6 in Figure 2?
16. What is IECs in line 360?
17. What is GPR43 in line 361?
18. What is RegIIIg in line 362?
19. What is Muc2 in line 365?
20. What is GI in line 378?
Please write bacterial species name with Italic in the section of REFERENCES.
Author Response
-Reviewer 3
Response: Thanks to the Reviewers/Editors for his/her appreciation of the manuscript.
1) Comment
This review described the diversity of microbiota under various factors including Helicobacter pylori infection but provides only poor impact for researchers. I think the authors had better reconstruct the contents in order to have further interested to this review. For example, the authors had better present the detailed microbiota altered by various factors using Tables and Figures. In addition, the authors had better indicate the Figures as for the expression mechanism of defensins via the MAPK/AP-1 signaling pathway concerned with the activation of NF-kB.
1) Response: Thank to reviewers for this suggestion that improves our manuscript. In order to clarify the function and the role of microbiota we have added two figures in the paragraph 3 and 6.5. Moreover to shed ligth on the activation of MAPK/AP-1 we have added an other figure in the paragraph 7.3.
2) Comment
The authors should alter H. Pylori to Helicobacter pylori(Italic) in TITLE.
2) Response: We have modified the text according the comment.
3)Comment
Please rewrite “belongin” to “belonging” in line 22 in ABSTRACT.
3) Response: Thanks the review for the suggestions. We have modified the text according the comment.
4) Comment
Please consider to alter “which expression.....” to “ in which the expression of b-defensins.......” in line 23 in ABSTRACT.
4)Response: We have modified the text according the comment.
5) Comment
Please consider to alter “researches” to “studies” in line 35 in INTRODUCTION.
Response: We have modified the text according the comment.
6. Comment
Does GC in line 60 mean gastric cancer?
Response: We have modified the text based on the comment. We used the full name specifying the acronym.
7) Comment
What is SCFA in line 178? Is it short-chain fatty acid?
7) Response: We have modified the text based on the comment. We used the full name specifying the acronym.
8) Comment
What is HFD in line 185?
8) Response: We have modified the text based on the comment. We used the full name specifying the acronym.
9)Comment
What is CRC in line 188?
9)Response: We have modified the text based on the comment. We used the full name specifying the acronym.
10) Comment
What is ABH in line 225?
10) Response: We have modified the text based on the comment. We used the full name specifying the acronym.
11) Comment
What is AMP in line 234?
11) Response: We have modified the text based on the comment. we used the full name specifying the acronym.
12) Comment
Please check whether there is a hyphen between HBD and number.
12) Response: it’s usually to insert the hyphen between HBD and number according Ensembl annotation.
13) Comment
What are IL-22 and ILCs in line 351?
13) Response: We have modified the text based on the comment. We used the full name specifying the acronym.
14) Comment
What is BD14 in line 352?
14) Response: We have modified the text based on the comment. We used the full name specifying the acronym.
15) Comment
What are AHR and mBD14 in line 353?
Response: We have modified the text based on the comment. We used the full name specifying the acronym
16) Comment
What is CCR6 in Figure 2?
16) Response: CCR6 is chemokine receptor. Chemokine receptors are part of the GPCR family (G-Protein Coupled Receptors).It is the receptor of defensins.
17) Comment
What is IECs in line 360?
17) Response: We have modified the text based on the comment. We used the full name specifying the acronym.
18) Comment
What is GPR43 in line 361?
18) Response: We have modified the text based on the comment. We used the full name specifying the acronym.
19) Comment
What is RegIIIg in line 362?
19) Response: We have modified the text based on the comment. We used the full name specifying the acronym.
20) Comment
What is Muc2 in line 365?
20) Response: We have modified the text based on the comment. We used the full name specifying the acronym.
21) Comment
What is GI in line 378?
21) Response: We have modified the text based on the comment. We used the full name specifying the acronym.
22) Comment
Please write bacterial species name with Italic in the section of REFERENCES.
22) Response: We have modified the text based on the comment.
Round 2
Reviewer 1 Report
Authors have addressed all my comments. However, I can see many typos yet. Please carefully revise this manuscript to amend this. For citing some of them: In figure 1: Bactiria, synethesis, metabolis; line 76: uman and many others.
Author Response
-Reviewer 1
Authors have addressed all my comments. However, I can see many typos yet. Please carefully revise this manuscript to amend this. For citing some of them:
Comments : In figure 1: Bactiria, synethesis, metabolis; line 76: uman and many others.
Responses: Thanks to the Reviewer for this suggestion that improves our manuscript.
· In figure 1 we have replaced the uncorrect words with bacteria, synthesis and metabolism.
· At line 76 we have replaced uman with human.
· Others typos: in figure 3 we have corrected the words boxed in green. At line 315 page 10 we have replaced the word PPARg with PPARg
Reviewer 3 Report
Dear Authors,
This review article became to be easy to understand by inserting the Table and Figures, and I got good responses from the authors.
Sincerely yours,
Reviewer 3
Author Response
Comment
Dear Authors,
This review article became to be easy to understand by inserting the Table and Figures, and I got good responses from the authors.
Sincerely yours
Response
Thanks to the Reviewer for his/her appreciation of the manuscript.